# IMPACT OF MOLECULAR REPRESENTATIONS ON DEEP LEARNING MODEL COMPARISONS IN DRUG RESPONSE PREDICTIONS

## ABSTRACT

Deep learning (DL) plays a crucial role in tackling the complexity and heterogeneity of cancer, particularly in predicting drug response. However, the effectiveness of these models is often hindered by inconsistent benchmarks and disparate data sources. To address the gaps in comparisons, we introduce CoMParison workflow for Cross Validation (CMP-CV), an automated cross-validation framework that trains multiple models with user-specified parameters and evaluation metrics. The effectiveness of DL models in predicting drug responses is closely tied to the methods used to represent drugs at the molecular level. In this contribution, we benchmarked commonly leveraged drug representations (graph, molecular descriptors, molecular fingerprints, and SMILES) to learn and understand the predictive capabilities of the models. We compare the ability of different drug representations to encode different structural properties of the drugs by using prediction errors made by models in different drug descriptor domains. We find that, in terms of the average prediction error over the entire test set, molecular descriptors and Morgan fingerprints perform slightly better than the others. However, we also observe that the rankings of the model performance vary in different regions over the descriptor space studied in this work, emphasizing the importance of domain-based model comparison when selecting a model for a specific application. Our efforts are part of CANcer Distributed Learning Environment (CANDLE), enhancing the model comparison capabilities in cancer research and driving the development of more effective strategies for drug response prediction and optimization.

## 1 INTRODUCTION

Cancer research is currently exploring innovative techniques to enhance treatment outcomes through the use of analytical models called Drug Response Prediction (DRP) models Yancovitz et al. (2012); Fisher et al. (2013); Adam et al. (2020). These models utilize machine learning (ML) and deep learning (DL) algorithms to forecast tumor responses to various drug treatments without the need for specific biomarkers. However, accurately predicting drug responses using ML and DL models is a critical challenge Baptista et al. (2020); Adam et al. (2020); Zuo et al. (2021). Each study typically develops custom model implementation and validation strategies, making it difficult to assess model capabilities across drug representation methods, architectures, and datasets Partin et al. (2023). With the increasing complexity of models and the diversity of datasets, there is a pressing need for robust methodologies to compare these models Park et al. (2023). However, the current landscape lacks consistency and standardization in terms of model comparison techniques. Traditional approaches often rely on performance scores from original publications, which leads to incomparable and inconsistent results. This hinders elucidating the precise factors that drive predictive performance. Therefore, it is crucial to establish a standardized and comprehensive comparison workflow to address the urgent need to understand drug representation and its impact on drug response prediction error.

In light of these challenges, we recently implemented the CoMParison workflow for Cross Validation (CMP-CV) - an automated cross-validation framework that enables simultaneous training and evaluation of multiple DL models using standardized datasets, preprocessing, and performance

metrics. CMP-CV provides infrastructure for controlled experimentation by systematically varying model hyperparameters and architectures. It also has built-in support for custom analytical functions, which facilitates deeper analysis of model representations and uncertainties.

When applying DRP models in real-world applications, such as predicting drug efficacy or identifying suitable cancer treatments, selecting the best model is crucial. While existing comparison methods utilize metrics like R2 (coefficient of determination), RMSE (Root Mean Squared Error), and AUC (Area Under the ROC Curve) to assess overall model accuracy, they fail to reveal critical information about each model's unique strengths and weaknesses. For instance, certain models might excel in specific domains of the drug descriptor space but be less accurate in other regions. In this work we analyze model performance within distinct domains of the drug descriptor space to identify the most effective models for specific drug candidates and determine if certain drug's molecular representations are superior to others. This type of analysis enables more informed decision-making when selecting a model for practical applications.

A significant challenge in drug response prediction is the lack of consensus on a suitable molecular representation, which is further complicated by the diversity of DRP models. Therefore, large-scale model comparison is necessary, and CMP-CV serves as a robust framework for this purpose. Its ability to accommodate user-defined Python functions to analyse model predictions allows for comprehensive benchmarking of models to determine the impact of various molecular representations on prediction errors. The current application of CMP-CV focuses mainly on comparing Cancer Drug Response Prediction (CDRP) models across diverse molecular descriptor spaces. This comprehensive comparison not only provides a deeper understanding of drug representation and its impact on drug response prediction errors but also highlights the relative strengths of various models on drug properties in different domains.

## 2 Results and Discussion

### 2.1 CMP-CV: Deep Learning Model Comparison Framework

The CANDLE/Supervisor framework (Wozniak et al., 2018) is a workflow application system designed for HPC infrastructure. Supervisor consists of multiple exemplar workflows, including simple sweeps, automated hyperparameter optimization, and other data analysis workloads. It is capable of calling into user-specified model codes via multiple techniques, including direct Python library invocation, shell command lines, and Linux container invocation. Supervisor coordinates these model executions via CANDLE "hyperparameters," which extend the notion of model training hyperparameters to include a range of other control variables. The hyperparameter set is standardized by the CANDLE Library (CANDLE Team, 2018).

The CMP-CV employs the Supervisor framework, which facilitates the integration of the containerized models described here along with their hyperparameters. Inside the workflow, depicted in Figure 1, a list of hyperparameter combinations is specified in an external file, encoded in a JSON format, and each training run is performed concurrently. In this manner, a very large HPC system can be efficiently used. Supervisor monitors training progress and keeps resources busy, almost eliminating the need for the workflow developer to consider concurrency. As each training run completes, a comparison function is invoked across the error metrics produced during training.

The CMP-CV system's unique integrated functionality offers a seamless process for analyzing prediction results, delivering comparable output metrics, and facilitating the integration of custom analytical functions, thereby providing users with a tailored analytical experience. One key feature that sets CMP-CV apart is its ability to accommodate user-defined Python functions, enabling users to seamlessly integrate custom analytical functions into the workflow. We utilised this capability to obtain drug response prediction errors in different regions in a drugs' molecular descriptor space. Our results highlight the importance of understanding where each model excels; this will enable us and the rest of the community to better leverage their predictive power in future applications.

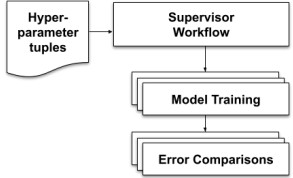

Figure 1: Architecture of the CMP-CV. The 'Error Comparisons' functionality contains python scripts to calculate the model errors corresponding to different regions in a drug's molecular descriptor space.

## 2.2 Overview of Drug Features and Representations

In the field of drug design and characterization, each drug is distinguished by a unique set of descriptors such as molecular structure, substructures, functionalities, physicochemical and biochemical properties, known targets, and clinical usage. These descriptors form the drug or molecular descriptor space. To apply machine learning techniques, it is necessary to create a numerical representation of these multifaceted descriptors. Investigating the effects of molecular representation on prediction accuracy provides valuable insights into current limitations of drug response modeling approaches. Our hypothesis is that the efficiency of a molecular representation depends on the model's ability to predict outcomes across various domains of the molecular descriptor space.

For instance, a molecular representation that includes fine details about ring structure can ensure good performance of the model, regardless of the number of rings in the drug molecule. It is important to mention that the model's performance variation for molecules with different numbers of rings is not solely due to its molecular representation strength. Other aspects of the molecule, such as molecular weight or number of atoms/hydrogen bonds can also change. However, if a model consistently fails to achieve good performance in a particular domain of the descriptor space, it indicates that the model's molecular representation is weak in that region.

## 2.3 Curated Existing Machine Learning Models for Comparison and Benchmarking

In our effort to understand the relationship between molecular representations and drug response predictions, we conducted a thorough curation and analysis of existing CDRP models, such as GraphDRP, DeepTTC, and HiDRA Nguyen et al. (2022); Jiang et al. (2022); Jin & Nam (2021). By applying CMP-CV to a standardized CTRPv2[1] dataset, we were able to compare and cross-validate these models, yielding important metrics that highlight their relative performance across the molecular descriptor space. This approach to curation and comparison represents a significant step towards enhancing the field of drug response prediction models.

Based on our literature survey on CDRP models Baptista et al. (2020); Partin et al. (2023), we identify that the models primarily use four categories of molecular representations: graph structures, SMILES encodings, Morgan fingerprints, and molecular descriptors. In Table 1, we list the CDRP models that leverage these distinct molecular representations. Our work focuses on comparing these four types of representations to understand their strengths and limitations.

To ensure a fair comparison of different drug representations, we also developed a model with the ability to switch between different molecular representations while using the same cell line representation. These models are hereafter referred to as **Graph**, **SMILES**, **Morgan** and **Descriptor**. More details about these models are given in the Appendix. Below is a brief description of the models from the literature.

**GraphDRP.** Nguyen et al. (2022) GraphDRP encodes drug molecules using graph convolutional layers followed by fully connected layers to arrive at a vector representation of length 128. The cell lines are initially represented using one hot encoding (735 dimensions). 1D convolutional operations followed by fully connected layers are used to convert the one hot encoded representation to a vector

---

[1]CSA Benchmark Datasets

Table 1: Models categorized based on the kind of drug representation they use

| Representation type | Models |
| --- | --- |
| Graph structure | SWnet (Zuo et al., 2021), DRPreter (Shin et al., 2022), GraphDRP Nguyen et al. (2022), DrugGCN(Kim et al., 2021) |
| SMILES encoding | DeepTTC Jiang et al. (2022), Paccmann Oskooei et al. (2019), tCNNS Liu et al. (2019) |
| Morgan fingerprints | DrugCell Kuenzi et al. (2020), HiDRA Jin & Nam (2021), DeepDSC Li et al. (2021), PathDSP Tang & Gottlieb (2021) |
| Molecular descriptors | CDRscan Chang et al. (2018), REFINED Bazgir et al. (2020), IGTD Zhu et al. (2021) |

of 128 elements. The drug and cell line representations are concatenated and fed through another fully connected neural network to arrive at the final prediction.

**DeepTTC.** Jiang et al. (2022) In DeepTTC, the SMILES string is tokenized using Explainable Substructure Partition Fingerprints (ESPF) Huang et al. (2019). The SMILES string is decomposed into multiple substructures and each substructure is assigned a number based on a provided vocabulary of substructures. This sequence of numbers is converted to a one-hot encoded matrix, and then transformed using a weight matrix. To this representation, a positional encoding is added to create the initial representation of the drug. This representation is sent through transformer encoder layers that contain multihead attention to arrive at the final drug representation.

**HiDRA.** Jin & Nam (2021) HiDRA is an attention-based model that aggregates gene expression data to drug fingerprint features to create a pathway-level network between the drug and cell line. The overall architecture is composed of four networks encompassing a drug, gene, and pathway level network followed by the response prediction network. Morgan fingerprints are used for drug representations and genes were grouped to pathways through the KEGG Pathway database to create the cell line feature. 4592 unique genes were used to create these features.

**ExtraTreesRegressor.** Geurts et al. (2006); Pedregosa et al. (2011) For the comparison, we also use an ExtraTreesRegressor model. This model is based on an ensemble of decision trees and does not utilize DL techniques. The model takes a simple concatenation of drug features and gene expression values of the cell lines as input.

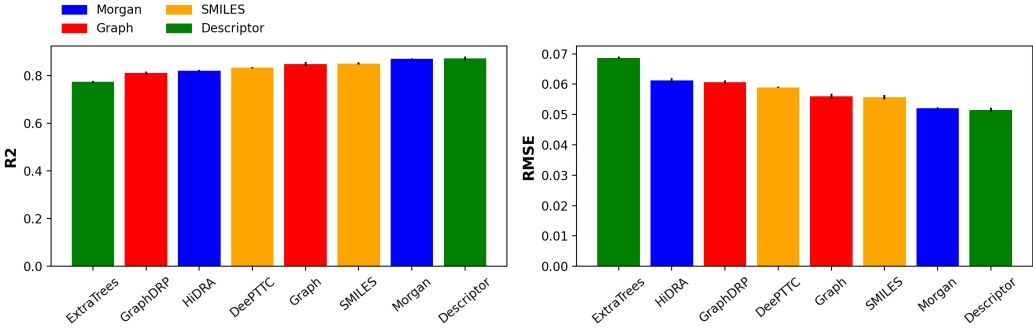

Figure 2: Comparative analysis of model prediction errors based on $AUC^{DR}$. Colors represent the type of representation used in each model.

## 2.4 Model Comparison

The CDRP models mentioned earlier were trained using the CTRPv2 dataset, which measures gene expression values in transcripts per million (TPM). These values were obtained from the CCLE DepMap[2] portal, while the response data were sourced from CTRP. As the dose-independent drug response metric, we use area under the dose response curve ($AUC^{DR}$). This $AUC^{DR}$ is what the

---

[2]https://depmap.org/portal/

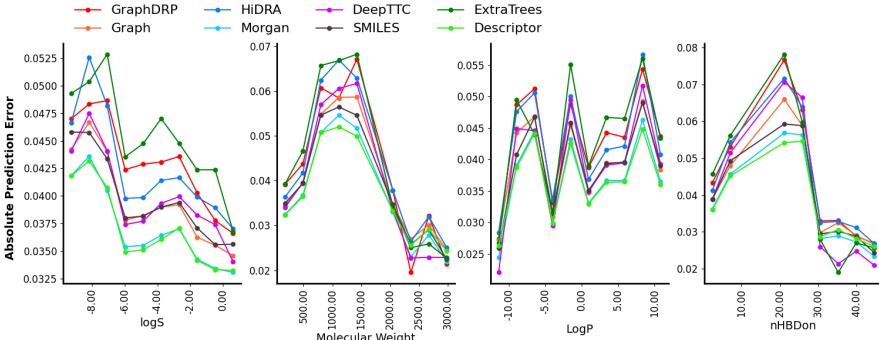

Figure 3: This figure presents a detailed analysis of AUC$^{\text{DR}}$ prediction errors in the domains of important drug properties such as logS, molecular weight, LogP, and nHBDon.

CDRP models attempt to predict. Further details on the dataset and model training are given in the Methods section.

The prediction accuracies for AUC$^{\text{DR}}$ are displayed in Figure 2. Based on the R2 results, it is observed that models utilizing molecular descriptors and Morgan fingerprints perform marginally better than the others. However, in this work, we aim to compare the performance of different models across various regions in the molecular descriptor space. To facilitate this comparison, we use Mordred Moriwaki et al. (2018) to generate molecular descriptors of the drugs. Descriptors that require three-dimensional coordinates were not taken into consideration. After obtaining the molecular descriptor values, they were divided into bins based on their ranges. These bins define the domains of the descriptors. Domain boundaries of continuous descriptors were found using NumPy[3]'s histogram function. Every unique value of a categorical descriptor was considered as a domain. A categorical descriptor is defined as one which consists of less than 20 unique integer values.

For instance, if a molecular descriptor value ranges from 5 to 95, to evaluate the performance of each model, we can group the molecules into intervals of 10 descriptor value units, such as 5-15, 15-25, and so on. This approach allows us to analyze a model's predictions in different regions in the descriptor space. In Figure 3, we present the variations in the AUC$^{\text{DR}}$ prediction error in the domains of solubility (logS), molecular weight, LogP, and the number of hydrogen bond donors (nHBDon), which are crucial descriptors in drug design Di & Kerns (2016). The information presented in Figure 3 offers two main advantages: Firstly, it increases the awareness of the users of these models regarding the limitations of the models in terms of the properties of the drug molecules. Secondly, it provides model developers with valuable insights into the deficiencies of their models.

### 2.4.1 EXPLORING DESCRIPTOR DOMAINS OF MODEL APPLICABILITY

Drug response prediction errors in the domains of logS, molecular weight, LogP, and nHBDon can significantly impact the performance of drug response prediction models. By identifying the domain errors of different models, we can determine which molecular descriptors have not been adequately represented in the model. This information can be used to enhance the performance of models by improving their representation in these descriptor domains.

Based on Figure 3, none of the ML models appear to perform well when the logS of the drugs is less than -7, and their errors decrease as the drug solubility increases. The Descriptor and Morgan models can be expected to perform best when predicting highly soluble drug candidates. These results facilitate the domain-wise representations comparison. For instance, in the high solubility regime (logS > 0), considering only the models with the same cell line representation, the goodness of the drug representation can be ranked as Morgan > Descriptor > Graph > SMILES.

In fact, one can construct a table showing the error-based model rankings for each domain as shown in Table (a), Figure 4. This resource empowers the systematic evaluation and determination of the

---

[3]NumPy

most efficacious models for drugs, characterized by distinct molecular attribute. For example, if we need to determine the best model for drugs with solubility varying in a wide range, the Descriptor model is the clear winner, followed by the Morgan model. For nHBDon however, the Descriptor model is more suitable when 2 > nHBDon < 8 (see Appendix Table 3). For drugs with over 35 hydrogen bond donors, DeepTTC is a superior model (Appendix Table 4) . These tables systematically categorize models based on their error rates within specific molecular descriptor domains, aiding in the seamless identification of the most adept models for predicting drug responses for drug candidates with particular molecular properties. Such information is useful for the robustness and reliability of drug response predictions.

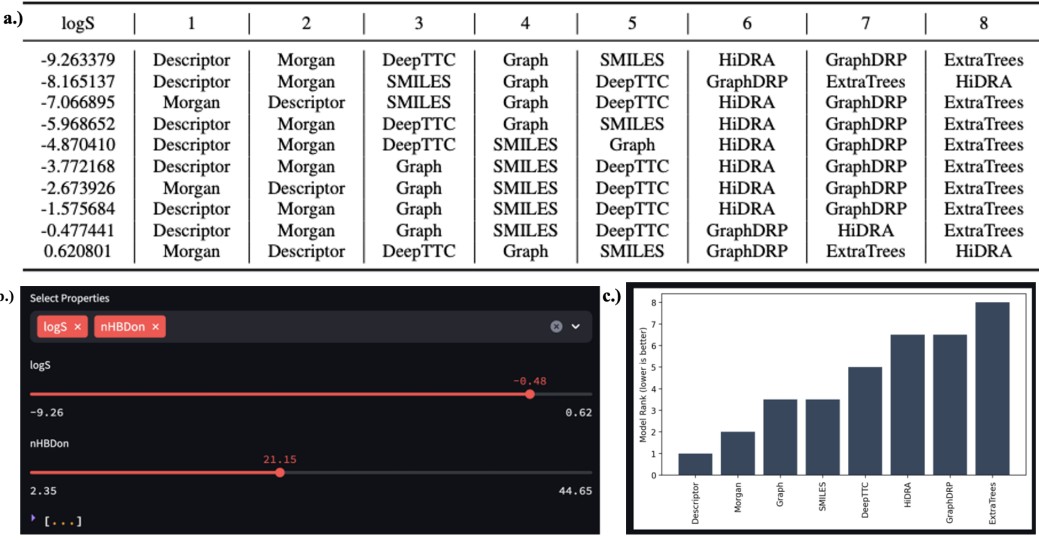

Figure 4: Table (a) systematically categorizes models based on their error rates within specific logS domains . Images (b) and (c) depict a web application that allows users to find model ranking based on multiple distinct molecular descriptor values. Values of more than 700 molecular descriptors can be changed (b) to obtain the corresponding model ranks (c).

We also designed a web application which allows a user to identify the models best suited for drug candidates described using multiple molecular descriptors. This interface allows the user to add as many as 786 molecular descriptors and adjust their values using the associated sliders. As shown in Figure 4 (b) and (c), once the descriptor values are chosen, a rank for each model is presented. These ranks are calculated by first looking up the model ranks corresponding to the chosen properties from tables similar to Figure 4, Table (a). If $n$ property values are selected, we have $n$ sets of model ranks. Each set contains ranks of $m$ models considered in the comparison. Next, the average rank of each model is found which is considered as the final model rank. Models are ranked from 1 to $m$, where 1 is the best rank and $m$ is the worst rank.

### 2.4.2 IDENTIFYING MODEL REPRESENTATION DEFICIENCIES

When dealing with over 1000 molecular descriptors, it can be challenging to determine which ones are most important for understanding how drug representation affects model performance. A logical assumption is that if a particular descriptor has been accurately encoded by a representation, then domain errors associated with that descriptor will be minimal. Conversely, if a representation fails to capture the intricate details of a molecular descriptor, domain errors corresponding to that descriptor will be significant.

We can determine the maximum error of a model for a specific domain. For instance, HiDRA has a maximum error of approximately 0.0525 at logS = -8 (Figure 3). These errors can be utilized to identify molecular descriptors that are not adequately encoded in the model's representation. This particular insight into individual errors per model can act as a pivotal tool for discerning molecular

descriptors that remain inadequately encoded within the model's architecture. Figure 5 displays the largest maximum and smallest minimum domain errors for each model, consisting of the top 5.

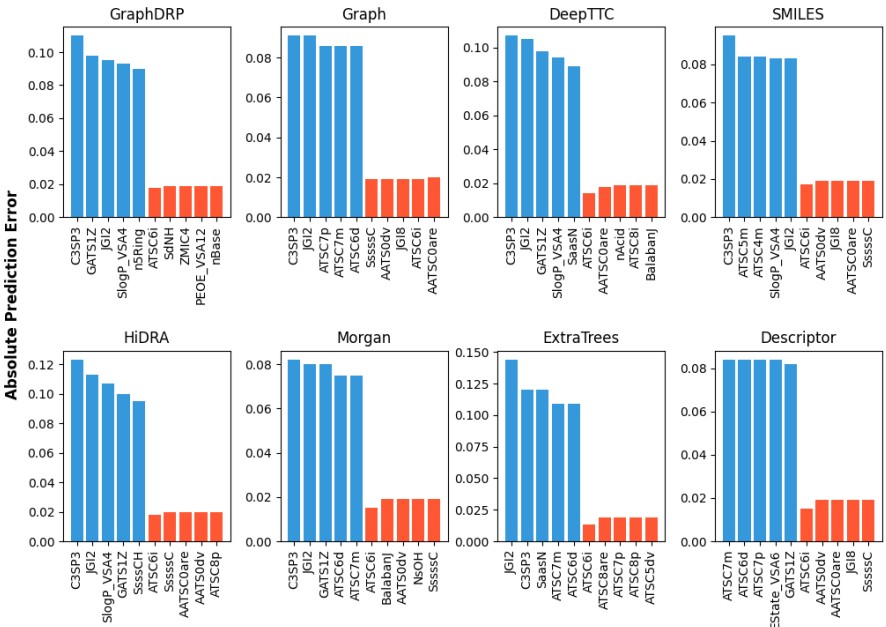

Figure 5: Descriptors that made maximum and minimum domain errors. Veritcal axis is the number of drug response values.

We notice that GATS1Z, C3SP3, SlogP_VSA4 and JGI2 are among the descriptors having the largest domain errors for most of the models. GATS1Z is the geary coefficient of lag 1 weighted by atomic number, C3SP3 is SP3 carbon bound to 3 other carbons, SlogP_VSA4 is a MOE type descriptor based on Wildman-Crippen LogP and surface area contribution, and JGI2 is the mean topological charge index of order 2 Moriwaki et al. (2018).

Table 2: Drug response prediction errors associated with $\text{AUC}^{\text{DR}}$ < 0.75 and $\text{AUC}^{\text{DR}}$ >= 0.75 cell-line – drug pairs.

|  | MAE | RMSE |
| --- | --- | --- |
| $\text{AUC}^{\text{DR}}$ < 0.75 | $0.06 \pm 0.003$ | $0.082 \pm 0.005$ |
| $\text{AUC}^{\text{DR}}$ >= 0.75 | $0.032 \pm 0.001$ | $0.044 \pm 0.001$ |

Figure 6 further demonstrates the error oscillations for the aforementioned descriptors, unfolding domains with the most significant errors: GATS1Z < 0.2, C3SP3 > 9, 50 > SlogP_VSA4 < 55, and JGI2 < 0.04. Such intricate data prove invaluable in decoding the root causes of subpar model performance and paves the path for consequential model enhancements. In fact, we notice that the prediction errors associated with $\text{AUC}^{\text{DR}}$<0.75 drugs are signicantly higher than those of $\text{AUC}^{\text{DR}}$>=0.75 drugs (see Table 2). In the Appendix, we investigate whether the error from the above descriptors is due to a common molecular structure motif or a deficiency of the representation.

Investigating further, observing drug response values ($\text{AUC}^{\text{DR}}$) in domains GATS1Z < 0.2 and GATS1Z > 1.5 (refer to Figure 7) reveals certain $\text{AUC}^{\text{DR}}$ values in the GATS1Z < 0.2 distribution do not originate from a densely populated region in the complete distribution. This correlation highlights the association of GATS1Z < 0.2 drugs with diminished drug response values.

In order to demonstrate how one can potentially use the information about domain errors to improve the model predictions, we pretrained the GraphDRP model to predict the molecular descriptors corresponding to largest error domains; GATS1Z, C3SP3, SlogP_VSA4, JGI2 and n5Ring. The

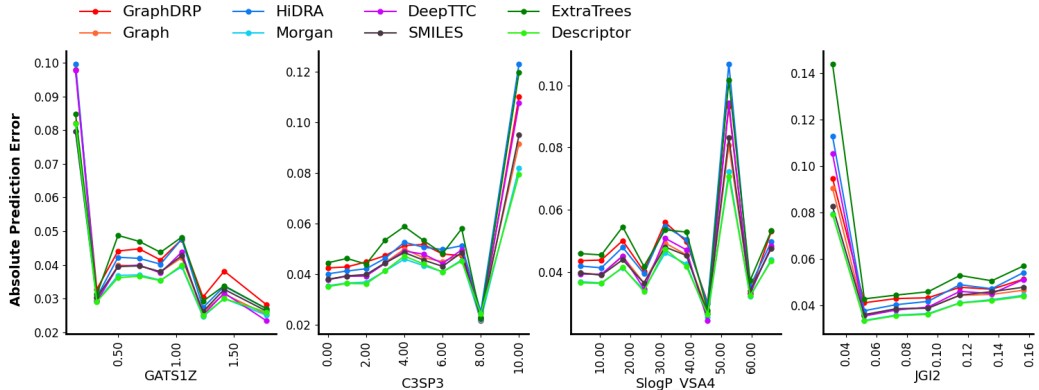

Figure 6: Visualization of error fluctuations within high-error descriptors domains. This plot is crucial for identifying and understanding the underlying causes of model performance

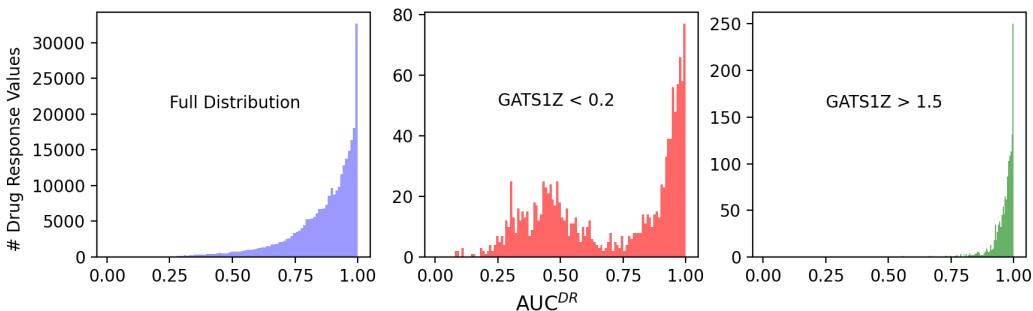

Figure 7: Examination of $AUC^{DR}$ distributions in different GATS1Z regions in the dataset.

pretraining GraphDRP model was created by replacing the last linear layer with three layers; one with three outputs for GATS1Z, SlogP_VSA4 and JGI2, another two with 11 and 8 outputs for unique values of C3SP3 and n5Ring respectively. The model was trained for 100 epochs with early stopping. After training, the weights of this model were loaded to the original GraphDRP model and trained for 100 epochs. Using the predictions of this model we obtained the domain errors again. Comparison of the logS, Molecular Weight, logP and nHBDon domain errors before and after pretraining are shown in Figure 8. We see significant error reductions in logS and LogP domains. We also observe a test set R2 improvement from 0.812 to 0.838 due to pretraining.

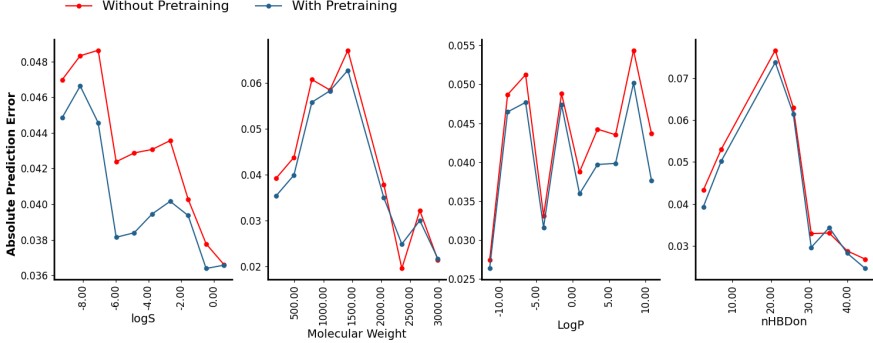

Figure 8: Reduction in GraphDRP error after pretraining.

## 3 METHODS

### 3.1 DATA AND MODEL TRAINING

The CTRPv2 dataset used in this work is from the CSA Benchmark Datasets curated as part of the IMPROVE[4] project. Cell line response data of this dataset were extracted from the Cancer Therapeutics Response Portal version 2. After extracting multi-dose viability data, a unified dose response fitting pipeline was used to calculate the dose-independent response metric, area under the dose response curve ($AUC^{DR}$). Drug data have been retrieved from PubChem (Kim et al., 2023). The CTRPv2 dataset has 720 cell lines and 494 unique drugs. The total number of drug response values is 286665.

The full dataset was divided into ten random train, validation, and test folds using different random seeds. This ensured that every drug-cancer cell combination was predicted at least once. The models were trained using the train set, the validation set is used for saving the best models. Except for the HiDRA model, others were trained for 100 epochs. As it takes about 30 minutes for a HiDRA epoch to complete, it was trained for 20 epochs. The predictions made by each of the test sets are recorded. These predictions are used to find the mean and the standard deviation of the prediction errors across the ten runs.

## 4 CONCLUSIONS

Domain error is a significant factor that can impact the performance of drug response prediction models. By utilizing our recently implemented CMP-CV framework and understanding the domain errors of different CDRP models, we can identify the molecular descriptors that have not been encoded with sufficient detail by the model's representation. This knowledge can be used to guide the selection of models for specific applications. We also introduce a web application which enables users to find the CDRP models better suited for drugs having specific molecular properties. We found that the prediction accuracy for drugs with a low solubility, particularly below the threshold logS < -7, dramatically decreases regardless of molcular representation. Increased drug solubility notably improves prediction accuracy with two models based on molecular descriptors and Morgan fingerprints preforming substantially better than other representation across the entire range for solubility. In addition, we can use the domain errors of models to improve the performance of models by focusing on improving their representation in these descriptor domains. Our analysis revealed that GATS1Z, C3SP3, SlogP_VSA4 and JGI2 are among the domains that might not be encoded with adequate detail by any of the molecular representations that could help improve the model prediction. By avoiding models with large errors in the domain of interest, we can obtain more reliable predictions from the models. We also show that using the descriptors corresponding to high-error domains as pretraining targets has a potential to improve model predictions.

In conclusion, molecular representation and feature domain exploration lays a robust foundation for not only recognizing and comprehending the domains contributing to the largest errors but also offers an opportunity for substantial model improvement.

---

[4]IMPROVE

## REPRODUCIBILITY STATEMENT

We have provided the instructions to run the CMP-CV and the code to perform the data analysis shown in the paper in the code.zip file.

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

# A APPENDIX

## A.1 MODEL RANKINGD FOR NHBDON DOMAIN.

How the model rankings change at different domains of nHBDon (number of hydrogen bond donors) space.

Table 3: Rankings of models for each nHBDon descriptor domain.

| nHBDon | 1 | 2 | 3 | 4 | 5 | 6 | 7 | 8 |
|--------|---|---|---|---|---|---|---|---|
| 2.350000 | Descriptor | Morgan | DeepTTC | SMILES | Graph | HiDRA | GraphDRP | ExtraTrees |
| 7.050000 | Descriptor | Morgan | Graph | SMILES | DeepTTC | GraphDRP | HiDRA | ExtraTrees |
| 11.750000 | GraphDRP | Graph | HiDRA | Morgan | DeepTTC | SMILES | ExtraTrees | Descriptor |
| 16.450000 | GraphDRP | Graph | HiDRA | Morgan | DeepTTC | SMILES | ExtraTrees | Descriptor |
| 21.150000 | Descriptor | Morgan | SMILES | Graph | DeepTTC | HiDRA | GraphDRP | ExtraTrees |
| 25.850000 | Descriptor | Morgan | SMILES | Graph | ExtraTrees | GraphDRP | HiDRA | DeepTTC |
| 30.550000 | DeepTTC | ExtraTrees | Morgan | Descriptor | SMILES | Graph | HiDRA | GraphDRP |
| 35.250000 | ExtraTrees | DeepTTC | Morgan | SMILES | Descriptor | Graph | HiDRA | GraphDRP |
| 39.950000 | DeepTTC | ExtraTrees | Morgan | Descriptor | GraphDRP | SMILES | Graph | HiDRA |
| 44.650000 | DeepTTC | Morgan | SMILES | ExtraTrees | Descriptor | Graph | HiDRA | GraphDRP |

Table 4: Average RMSE values for DeepTTC and Descriptor models in two regions in the nHBDon space.

| nHBDon | DeepTTC | Descriptor |
|--------|---------|------------|
| $< 8$ | $0.0587 \pm 0.0004$ | $0.0513 \pm 0.0015$ |
| $> 35$ | $0.0336 \pm 0.0044$ | $0.0385 \pm 0.0006$ |

## A.2 SUPERVIOSOR FRAMEWORK

### SUPERVISOR FRAMEWORK SCALABILITY

Supervisor was designed as an Exascale Computing Project application, meaning it was designed from the beginnning for exascale computers. Supervisor is built around the Swift/T Wozniak et al. (2013); Armstrong et al. (2014) workflow language and runtime. Swift/T is an MPI-based workflow system, so communication for task distribution and monitoring is performed over MPI Forum (1994), the messaging layer provided by machine vendors for efficient use of large-scale computers. Swift/T is scalable through two architectural innovations. First, the task distribution is coordinated by a network of multiple task servers, which use an efficient work-stealing algorithm to distribute work. Secondly, the control logic itself generated from the user workflow script is evaluated over this distributed fabric, meaning that the workflow evaluation itself is also scalable.

### SUPERVISOR USABILITY FOR DEEP LEARNING WORKFLOWS

Supervisor has many usability features for deep learning applications. These are based on features of the Swift/T language and the Supervisor scripts that wrap the core workflow features with easier to use scripts for launching workflows. For example, Swift/T contains multiple mechanisms for calling back into user code through Python interfaces, command lines, and other languages like Tcl and R Wozniak et al. (2015). Supervisor is launched with the `supervisor` tool, which accepts a workflow name, site specification, and configuration file. The workflow name is essentially a label to the workflow, such as "`CMP-CV`" for the present case. The site specification contains settings for the computing system in question, such as program locations for Swift/T, Python, etc. The configuration file contains any additional settings, such as scheduler items including walltime, resources to allocate, parameters for a workflow control algorithm in use, etc.

Internally, Supervisor contains scripts to glue the workflow system to user models through the "model shell." For the CMP-CV case, this script sets up the container for execution, handles the hyperparameters, finds and runs the container with its standard command line, and collects results. Everything here is logged into a per-model log for human examination and possible debugging later.

## A.3 GRAPH, SMILES, MORGAN AND DESCRIPTOR MODELS

This section contains details on the **Graph**, **SMILES**, **Morgan** and **Descriptor** models introduced in the Section 2.3. These four models use the same cell-line representation but different drug representations. The cell-line representation is created by feeding the 1007 gene expression values to a fully connected neural network. The drug representation of the **Graph** model is created using a

graph neural network Panapitiya et al. (2022) consisting of graph convolutional layers. In the **Morgan** model, a drug molecule is represented using a Morgan fingerprint in the form of a bit vector of size 1024. RDKit[5] is used for this fingerprint generation. The drug representation of the **SMILES** model is created as it is done in the DeepTTC[6]Jiang et al. (2022) model. The drug representation of the **Descriptor** model is initialized using 783 molecular descriptors generated using the Mordred package Moriwaki et al. (2018). These descriptors are fed into a fully connected neural network to create the final drug representation.

## A.4 Unraveling the Role of Molecular Structure in Drug Error

By learning from the feature domain, we explored the potential relationships between the drug structures and their corresponding features. In Figure 9, a visualization technique is employed to embed the circular Morgan fingerprint representations of the drugs, utilizing UMAP (McInnes et al., 2018). This method allows for the reduction of high-dimensional (2048-bit) fingerprint vectors into an accessible two-dimensional representation. Subsequently, the desired descriptor values were overlaid utilizing a color spectrum.

Upon close scrutiny of the four plotted graphs covering GATS1Z, SlogP_VSA4, C3SP3, and JGI2, an identification of the regions of chemical space encompassed by the data is unveiled. This visualization serves as a tool, highlighting the specific regions of space each descriptor predominantly occupies, offering an insightful glance into the diverse chemical territories. From this figure we see that there is close clustering for the $50 < S\,\mathrm{logP\_VSA4} < 55$ and $\mathrm{JGI2} < 0.04$ molecules, highlighted with red X's. This infers that the high error drugs in these descriptor domains exhibit similar structural motifs that possibly contribute to the error. The opposite is also true where the descriptors $\mathrm{GATS1Z} < 0.2$ and $\mathrm{C3SP3} > 9$ show sparser data points. This points to these descriptors being less correlated with certain similar structural motifs. This embedding offers yet another way to utilize the results gathered above to draw conclusions about a model's weaknesses and strengths. A closer look at examples of these structures can be found in Figure 10 and Figure 11.

---

[5]RDKit

[6]DeepTTC

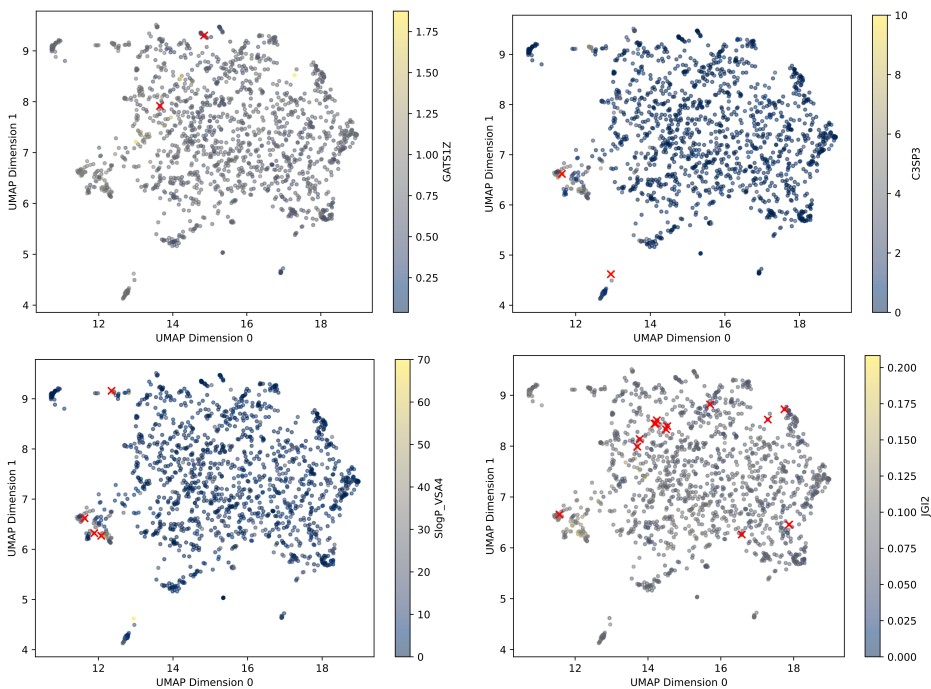

Figure 9: UMAP embeding of molecular fingerprints with overlay of molecular features of interest: GATS1Z, C3SP3, SlogP_VSA4, and JGI2. The color of each point corresponds with it's associated value and the red X's highlight the molecules identified as having the highest error from Figure 5.

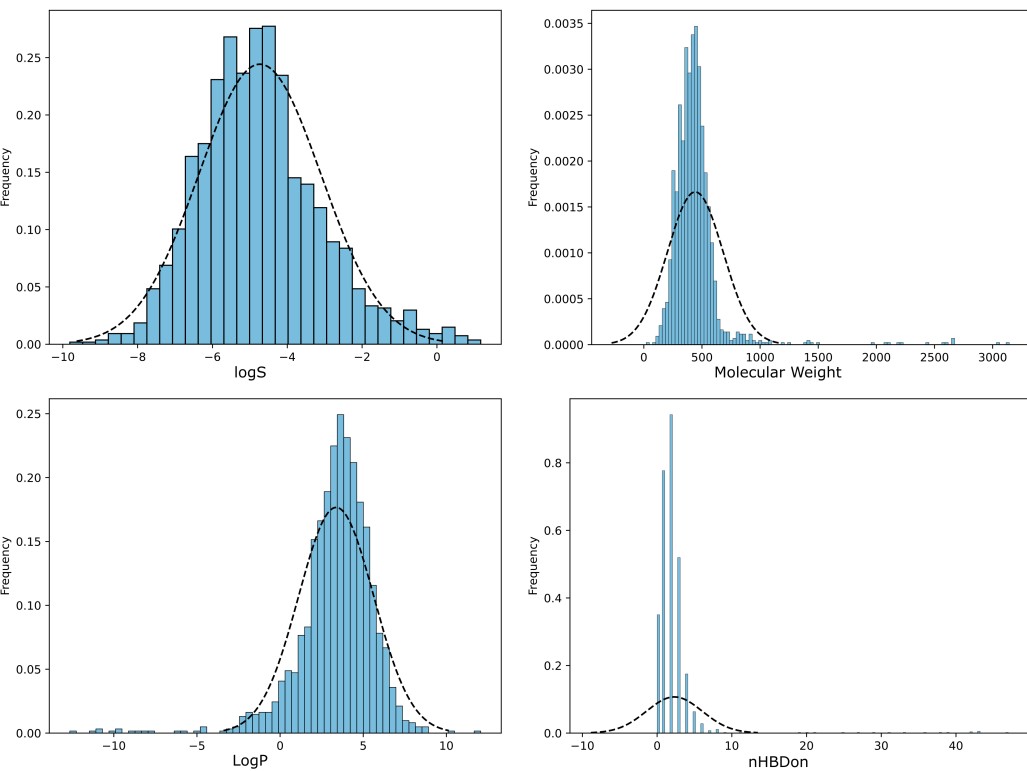

Figure 10: Distributions of drug like properties over the used dataset. Covers solubility (logS), Molecular Weight, the partition coefficient (LogP), and number of Hydrogen donors (nHBDon).

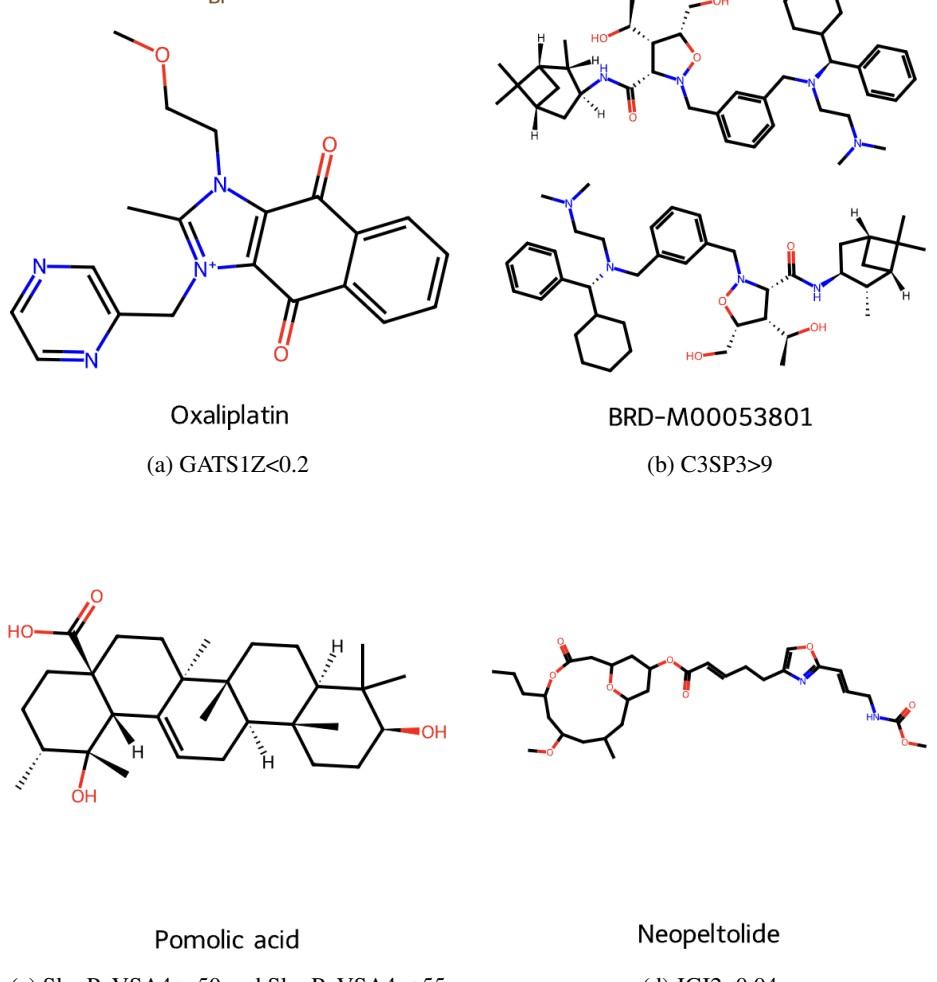

Oxaliplatin

(a) GATS1Z<0.2

BRD-M00053801

(b) C3SP3>9

Pomolic acid

(c) SlogP_VSA4 > 50 and SlogP_VSA4 < 55

Neopeltolide

(d) JGI2<0.04

Figure 11: Example drug molecules in GATS1Z, C3SP3, SlogP_VSA4 and JGI2 domains.

