# OpenReview forum: "Impact of Molecular Representations on Deep Learning Model Comparisons in Drug Response Predictions"
_ICLR.cc/2024/Conference — Submitted to ICLR 2024_

### Official Review · Reviewer_jtTX · 2023-10-29

**Soundness:** 2 fair
**Presentation:** 2 fair
**Contribution:** 1 poor
**Rating:** 3
**Confidence:** 4

**Summary:**

Using the CTRPv2 dataset and a training orchestration system called CMP-CV, the authors train different deep learning architectures using different molecular representations, then perform an exploratory data analysis on the errors those models make, slicing up the errors by different molecular properties. They show that the error distributions are non-uniform across these properties.

**Strengths:**

- The authors call to attention to the fact that predictive error is typically non-uniform.

**Weaknesses:**

- This paper is primarily an EDA of the errors of trained models, and does not rise to the level of making a contribution significant enough for a main track paper, and is better suited for a workshop.
- The authors use too much space describing CMP-CV, which appears to be a standard job orchestrator, and does not rise to the level of making a contribution. More writing should be dedicated to describing how the metrics were actually computed (see Questions).
- The authors point out the non-uniformity of errors, but do not provide actionable recommendations on how one ought to proceed with this knowledge.
- R2, RMSE and MAE are correlated, no need to show them all in Figure 1.
- The trends for all models in Figures 3 and 4 are roughly the same, suggesting that the non-uniformity in prediction error is due more to the dataset than any choice of architecture or molecular representation.
- The UMAPs of Figure 6 are very uninformative, and do not appear to support the author's point, unless there are many red Xs at each point in the space. The point of this figure could be expressed very differently, perhaps by tanimoto similarities of these clusters compared to average similarity or something like that.
- The point of Tables 2 and 3 could be expressed in just a few lines.
- The paper says that 10 models were trained on 10 random train/val/test splits, which is not standard practice - the test set is usually fixed across all CV splits.

**Questions:**

- many of the details of how metrics were obtained are left out, e.g.
  - What exactly does each model predict? It seems to be gene expression values, but it is not clear in the paper
  - If expression values, then how do the authors get to single R2, RMSE, and MAE values in Figure 1? The line "This figure delineates the areas where each model exhibited the highest number of errors" is not clear.
- How were the bins of Tables 2 and 3 chosen?

---

> ### Author Response · Authors · 2023-11-22
> **Response to Reviewer jtTX**
>
> Thank you so much for reviewing our work and for the valuable comments. Please note that the changes we made in the revised paper are temporally highlighted in blue. Also, Figure and Table number have changed in the revised version.
>
> - About the contribution of this work (weakness 1):
> In this work, we use errors made by models in different regions of the molecular descriptor space to compare drug/molecule representations used by different models. In the revised version, we have also shown how this knowledge on domain-based errors can be used to improve model accuracies (section 2.4.2). Also, we introduce a web application that allows users to find models better suited for drugs having specific molecular properties (section 2.4.1).
>
> - About CMP-CV (weakness 2):
> We rewrote the sections pertaining to CMP/CV to better describe it’s role (section 2.1). The main contribution of this work is to compare drug representations used by models in terms of model errors in different regions in the molecular descriptor space and provide insight on the limitations of each representation. In the revised version we have clarified the prediction target (section 2.4, lines 153-154).
>
> - About recommendations (weakness 3):
> As mentioned in the response to weakness 1, in the revised version, we demonstrate how this knowledge can be used to improve model accuracies (section 2.4.2).
>
> - About Figure 1 (Figure 2 in the revised version)(weakness 4):
> In the revised version, we provide only R2 and RMSE results (Figure 2) as these are the most popular metrics used by the cancer drug respones prediction models.
>
> - About the impact of molecular representation (weakness 5):
> It is true that the trends for different models are similar. However, we also notice that magnitude of the model errors in a given descriptor domain vary across different models. We hypothesize that one reason for this variation is the differences in the expressiveness of the representations used in different models.
>
> - About UMAP results (weakness 6):
> We feel the UMAP is simply a visualization that highlights whether there is a structurally relevant reason as to why there may be high error for a given feature. So, if a large cluster of red X’s appeared on the UMAP, there would be reason to believe that the high error could be due to the structure but if there are a lot of spread-out red X’s, the given error is not necessarily related to the structure. However, we agree this wasn’t described well in the paper and have chosen to move it to the appendix as it does not provide the take home message intended.
>
> - About Table 2 and 3 (weakness 7):
> We agree that the main takeaways of Table 2 and 3 can be expressed using sentences. In the revised version we keep only Table 2 in the main text (now part of Figure 4) as it makes it easier for the reader to clearly see the changes in models’ rankings in different regions of the descriptor space. We moved Table 3 and 4 to the Appendix.
>
> - About test set (weakness 8):
> For a given train/validation/test split, all the models used same train, validation and test data. We wanted to make predictions using all the data. This is why we split the dataset into train, validation and test folds in a cross validated manner. Had we used a fixed test set our results would only correspond to a smaller subset of the drug-cell line combinations, thus limiting the validity of our conclusions.
>
>
> - About the prediction target (Question 1):
>
>    - (Question 1.1) “What exactly does each model predict?” :
>
>    - Answer: We predict the drug response value, which is quantified by the “Area Under the Dose Response Curve”. We had mentioned this in lines 152-153. In the revised version we also added the sentence “This AUCDR is what the CDRP models attempt to predict” to further clarify what the models predict (lines 153-154).
>
>    - (Question 1.2) “If expression values, then how do the authors get to single R2, RMSE, and MAE values in Figure 1? The line "This figure delineates the areas where each model exhibited the highest number of errors" is not clear.”:
>
>    - Answer: As mentioned above we predict one drug response value per drug -cell line combination. Thus, we get a single values for R2, RMSE and MAE.
>
> - About defining the bins (Question 2):
>
>     - Answer: In the revised version, we added a description on how the bins were found (lines 162-166). “These bins define the domains of the descriptors. Domain boundaries of continuous descriptors were found using NumPy’s histogram function. Every unique value of a categorical descriptor was considered as a domain. A categorical descriptor is defined as one which consists of less than 20 unique integer values.”

---

### Official Review · Reviewer_69P5 · 2023-10-31

**Soundness:** 2 fair
**Presentation:** 3 good
**Contribution:** 2 fair
**Rating:** 3
**Confidence:** 4

**Summary:**

This paper presents an empirical study aimed at analyzing the performance of existing deep learning models for drug response prediction. The authors introduce CMP-CV, a framework for cross-validating multiple deep learning models using user-specified parameters and evaluation metrics. This study utilizes the CTRPv2 dataset to compare eight models across four different molecule representations. The experimental results highlight the significant impact of molecular representation on the prediction performance of deep learning models for drug response prediction.

**Strengths:**

- This paper addresses an important application of machine learning in drug discovery.

- The authors provide the code necessary for reproducing the experiments.

**Weaknesses:**

- While the paper offers some insights into existing machine learning models, its technical novelty within the machine learning context is somewhat restricted. In particular, its primary contribution lies in error analysis to find which areas in the drug space the ML models does not achieve good performances rather than shedding a light on developing novel methods to enhance prediction performance for drug response. This aspect may fall short of the acceptance criteria for ICLR.

- The conclusion that molecular representation significantly influences drug response prediction performance appears to be straightforward and lacks novel insights for the ML-based drug discovery community.

**Questions:**

Please see the Weaknesses.

---

> ### Author Response · Authors · 2023-11-22
> **Response to Reviewer 69P5**
>
> Thank you so much for reviewing our work and for the valuable comments. Please note that the changes we made in the revised paper are temporally highlighted in blue. Also, Figure and Table number have changed in the revised version.
>
> The major goal of this work is to create a framework to understand the limitations in the drug representations used by models. The users of this framework use this kind of understanding to improve the models. In the revised version of the paper, we have provided an example for how to potentially use domain errors to improve model predictions (section 2.4.2). We pretrained the GraphDRP model to predict molecular descriptors corresponding to largest domain errors and used these pretrained weights to retrain a drug response prediction model. We find that this strategy improves both R2 and reduces domain-based errors.
>
> Also, in this work, we provide more information about which components of the drug representation could affect the prediction accuracies of the models studied in this work. As far as we know, ours is the first work that analyses model prediction errors in terms of domains in the drug descriptor space. In the revised version, we also introduce a web application that allows users to find models better suited for drugs having specific molecular properties (section 2.4.1).

---

### Official Review · Reviewer_gDji · 2023-11-06

**Soundness:** 1 poor
**Presentation:** 1 poor
**Contribution:** 1 poor
**Rating:** 1
**Confidence:** 2

**Summary:**

Certain models for drug response predictions favour certain drug representation, and this is a recurring problem in feature-based drug prediction tasks. This paper studies the inductive bias of drug representation in drug response prediction tasks.

The authors demonstrate that molecular descriptors and SMILES strings are effective drug representations for drug response prediction tasks.

**Strengths:**

The problem of studying the inductive biases for drug response prediction is interesting and relevant in drug discovery. I found the analysis on the effectiveness of certain representations across drug domains interesting, for example, the takeaway that the descriptor and morgan representation models are more effective with highly soluble drug candidates is neat. This includes the importance measure of molecular descriptors that are tightly captured by certain representations.

**Weaknesses:**

The biggest weakness of this work is the writing. I found the writing to be very difficult to follow. Couple points:

- Why is the method (CMP-CV) left to the end of the paper after the results?
- It required a few reads to disambiguate between feature space, representation, molecular descriptors, drug domains, and drug regions. I think in the next iteration of this work, time needs to be invested to expand on the different terminology used in this paper.
- In Figure 1, why are the duplicates in the x-axis and the legend?
- In Page 4, section 2.3.1, paragraph 2, is it not rather that the ML model appears to perform better when the log S of the drug is *more* than -7 ?
- In Figure 1&2, why do refer to the Area Under the Roc Curve (AUC), when the results are for the R2, RMSE and MAE ?

I am also still unsure what is the CMP-CV workflow exactly. It seems to be more of an engineering effort, that is largely handled by the CANDLE framework?

**Questions:**

Could the authors help me understand what is the key contribution of the CMP-CV workflow? It appears to be a hyperparameter sweep that is commonly used to evaluate machine learning models.

---

> ### Author Response · Authors · 2023-11-22
> **Response to Reviewer gDji**
>
> Thank you so much for reviewing our work and for the valuable comments. Please note that the changes we made in the revised paper are temporally highlighted in blue. Also, Figure and Table number have changed in the revised version.
>
> - About (CMP-CV) left to the end (weakness 1):
> We agree that there is some confusion regarding how we presented the details on CMP/CV. In the revised version, we provided a clearer description about how CMP/CV facilitates large scale domain-based model/representation comparison (section 2.1).
>
> - About feature space, representation, molecular descriptors, drug domains, and drug regions (weakness 2):
> We improved the introduction of terminology in the revised version (lines 92-95 and 162-166). The term ‘feature’ is replaced with ‘descriptor’. ‘Drug domains’ and ‘drug regions’ are replaced with ‘descriptor domains’. A drug molecule can be represented using molecular descriptors. The space formed by a set of descriptors (descriptor1, descriptor2, descriptor3,…. ) is called the descriptor space. Each molecular descriptor can be divided into domains defined by two boundary values (lower and upper limits).
>
> - About duplicates in the x-axis (weakness 3):
> Each image in Figure 1 (now Figure 2) shows performance of the same 8 models in terms of three different metrics, R2, RMSE and MAE. The color codes in the legend represent the molecular representation used in each of the 8 models. This is why each color is repeated in each image. For example, ‘Morgan’ and ‘HiDRA’ models use Morgan fingerprints to represent the drug molecule. Thus, a single color (blue) is used to color the bars corresponding to ‘Morgan’ and ‘HiDRA’ models accuracies. We understand that this could be little confusing. We added a sentence in the figure caption clarifying the meaning of the legend colors.
>
> - About page 4, section 2.3.1, paragraph 2 (weakness 4):
> We have mentioned that “none of the ML models appear to perform well when the logS of the drugs is less than -7”. This fact is true. The models’ errors are largest in the logS <-7 region. We have also stated that “their errors decrease as the drug solubility increases”, which I think is similar to what the reviewer tries to point out here.
>
> - About Area Under the Roc Curve (AUC) (weakness 5):
> R2, RMSE and MAE were calculated using the predicted and actual drug response value. The drug response value in this case is Area Under the dose response Curve. We notice we have used AUC in two different contexts in the text. One is to denote area under the dose response curve. And the other is to denote the general classification metric area under the roc curve. This is a mistake. In the revised version, we used AUCDR to denote the area under the dose response curve.
>
> - About CMP-CV workflow (weakness 6 and question 1):
> Addressed in the response to weakness 1.

---

### Official Review · Reviewer_Pfg9 · 2023-11-09

**Soundness:** 2 fair
**Presentation:** 2 fair
**Contribution:** 2 fair
**Rating:** 5
**Confidence:** 3

**Summary:**

This paper presents an automated cross-validation framework for drug response, coined CMP-CV, which trains multiple models with user-specified parameters and evaluation metrics. To achieve this, this paper benchmarked the commonly utilized drug representations (graph, molecular descriptors, molecular finger prints, and SMILES) on the proposed CMP-CV. The authors analyzed the results in various evaluation metrics, including average prediction error.

**Strengths:**

- The paper is well-written and easy to understand.
- The paper studies an important task of ML for drug discovery, which suffers from the lack of general benchmarks for evaluation.

**Weaknesses:**

- Unclear contribution: The paper compares existing methods based on the already proposed dataset. The evaluation metrics are also common, e.g., AUC, prediction error.  It is unclear which parts are the main contribution of this work.

- Insufficient analysis: Although figure 2 and table 2,3,4 seem interesting, the paper only presents the results without analysis, e.g., hypothesis or justification.

- Lack of descriptions about core technique: This paper repeatedly refer CANDLE framework. However, the description in Section 3.2 does not provide sufficient information to understand the framework.

- Lack of comparison with other benchmarks: There are several benchmarks for drug discovery, e.g., [1], [2]. However, there is no comparison about those works.

[1] Wu et al., MoleculeNet: a Benchmark for Molecular Machine Learning, Chemical Science 2018\
[2] Stanley et al., FS-Mol A Few-Shot Learning Dataset of Molecules, NeurIPS 2021

**Questions:**

- What does the overlapped vertical bar mean? (in Figure 3, Morgan-ATSC7p and ExtraTrees-ATSC7p)

- Please provide the more description of CANDLE framework and the contribution of this paper upon (or based on) the CANDLE framework.

- What is the main novelty of this work? In other words, what was the main difficulty to make this benchmark? Isn't this work a simple combination of existing methods, dataset, and evaluation metrics?

- What is the main advantage of this benchmark, compared to prior benchmarks [1,2] for drug discovery?

---

> ### Author Response · Authors · 2023-11-22
> **Response to Reviewer Pfg9**
>
> Thank you so much for reviewing our work and for the valuable comments. Please note that the changes we made in the revised paper are temporally highlighted in blue. Please note that Figure and Table number have changed in the revised version.
>
> - About Unclear contribution (weakness 1):
> We think the reason for this confusion could be Figure 2 (Figure 1 in the old version). Figure 2 is to depict how model comparison is typically carried out using the metrics based on the average prediction errors across all the data points in the test set.  However, our objective in this work is to perform an extended model comparison in terms of prediction errors in different domains of the drug descriptor space. We have mentioned this fact in lines 47-55 and 158-159. In the revised version, we also modified text to clarify the role of Figure 2 results (lines 156-159). Our hypothesis is that, since a drug can be represented in terms molecular descriptors, the errors in different regions of the molecular descriptor space reflect strengths and weaknesses of the drug representation. Our goal is to compare models in terms of these strengths/weaknesses of the drug representation.
>
> - About Insufficient analysis (weakness 2):
> As mentioned in the previous note, our hypothesis is that a drug can be represented in terms of molecular descriptors therefore enabling us to compare models in terms of their prediction errors in different regions in the molecular descriptor space. We agree that we have not provided sufficient justification. In the revised version of the paper, we provide details how the molecular descriptors corresponding to largest domain errors can be potentially used to pretrain models and thereby to reduce the prediction errors (section 2.4.2, lines 238-248).
>
> - About Lack of descriptions (weakness 3):
> We agree that sufficient information about the CANDLE framework is not provided in the main text. This is mainly due to the page limit. In the revised version, we described CANDLE and CMP/CV in more detail in the main text (section 2.1).
>
> - About Lack of comparison (weakness 4):
> Our goal of this work is to introduce an alternate model comparison method for which we used drug response prediction as a test case. We agree that carrying out our analysis using more datasets is important. However, such analysis for datasets in Moleculenet cannot be carried out using the drug response prediction models we have used in this work. In a future work, we plan to extend this work to perform our model/representation comparison using several benchmark datasets including Moleculenet.
>
> - About overlapped bars (question 1):
> Figure 3 shows descriptors corresponding to largest and smallest drug response prediction errors in descriptor domains. The prediction errors of a model vary across different regions of a given descriptor. For Morgan model, ATSC7p descriptor’s domain errors are among the largest domain errors and also among the smallest domain errors. In the revised version, to avoid confusion, we removed the overlapping bars from Figure 4 (now Figure 5).
>
>
> - About CANDLE (question 2):
> Please refer section 2.1
>
> - About novelty (question 3):
> The novelty of this work is the method to compare models and molecular representations using the domain errors in the molecular descriptor space; not just using the averaged errors like RMSE and MAE. Using CMP/CV, we provide an automated framework to do this comparison for different hyperparameter combinations (In the revised version we have added more clarification on the role of CMP-CV (section 2.1)). The knowledge on such domain-based errors facilitates decision making regarding when and how to use models. For example, according to logS graph in Figure 3, we find that none of the models are suitable to make predictions when very low solubility (logS < -7) drugs are involved.
>
> As we now know certain representations are worse in the low solubility regime than other, this kind of information also facilitates coming up with improvements to models’ representations. In the revised paper, we have introduced our web application that can rank models based on drug properties selected by a user (section 2.4.1, lines 199-207).
>
> - Main advantage (question 4):
> As mentioned in the previous note, the main advantage is that we get more insight about a model’s drug representation in terms of different molecular descriptors. This can lead to model improvements. Traditional comparison metrics like R2, RMSE do not provide such insights. (first mentioned in lines 47-55)

---

> ### Comment · Reviewer_Pfg9 · 2023-11-23
> **Thank you for the rebuttal.**
>
> Thank the authors for their reply. In this moment, I still have some remaining concerns.
>
> ---
> [About unclear contribution, W1]: I agree with the reviewer 69P5 and jtTX, which pointed out that the main contribution of this work, i.e., error analysis, is not significant for a main track paper.
>
> ---
> [About lack of descriptions, W3]: While I appreciate the authors' efforts to revise their manuscript, some terminologies are still vague. For example, what is HPC infrastructure? How are they constructed? Does the authors implemented it? or borrowed from existing framework?
>
> ---
> [About lack of comparison, W4]: I partly agree that MoleculeNet cannot be carried out using the drug response prediction models. Then, is this paper the first work to benchmark drug response prediction models?

---

### Meta-Review · Area_Chair_qACy · 2023-12-05

**Metareview:**

This paper proposes CMP-CV, an automated cross-validation framework designed for drug response prediction. The framework trains multiple models using user-specified parameters and evaluation metrics. The authors benchmarked commonly used drug representations such as graphs, molecular descriptors, molecular fingerprints, and SMILES within the CMP-CV framework. However, all four reviewers are negative about accepting the paper. After careful review of the paper and response, AC also thinks that the author had not provided sufficient arguments to convince the reviewers. AC acknowledges that CMP-CV is a tool for evaluating and comparing the effectiveness of different drug response prediction models in a systematic way, but acknowledges that the technical novelty in the context of machine learning is somewhat limited, as noted by all reviewers. Developing new methods to improve prediction performance for drug response, as well as error analysis, may alleviate concerns related to unclear contribution.

**Justification For Why Not Higher Score:**

All five reviewers (including AC) are negative for the acceptance of this paper.

**Justification For Why Not Lower Score:**

N/A

---

### Decision · Program_Chairs · 2024-01-16

Reject